# Mobile Device-Based Video Screening for Infant Head Lag: An Exploratory Study

**DOI:** 10.3390/children10071239

**Published:** 2023-07-18

**Authors:** Hao-Wei Chung, Che-Kuei Chang, Tzu-Hsiu Huang, Li-Chiou Chen, Hsiu-Lin Chen, Shu-Ting Yang, Chien-Chih Chen, Kuochen Wang

**Affiliations:** 1Department of Pediatrics, Kaohsiung Medical University Chung Ho Memorial Hospital, Kaohsiung Medical University, Kaohsiung 807, Taiwan; 1000461@gap.kmu.edu.tw (H.-W.C.); ch840062@cc.kmu.edu.tw (H.-L.C.); staceyyang0906@gmail.com (S.-T.Y.); 2Department of Biological Science and Technology, National Yang Ming Chiao-Tung University, Hsinchu 300, Taiwan; 3Department of Pediatrics, Kaohsiung Municipal Siaogang Hospital, Kaohsiung, Kaohsiung Medical University, Kaohsiung 812, Taiwan; 4Department of Computer Science, National Yang Ming Chiao Tung University, Hsinchu 300, Taiwan; ckchang0424.cs09@nycu.edu.tw; 5Department of Rehabilitation Medicine, Kaohsiung Medical University Hospital, Kaohsiung 807, Taiwan; tzu-hsiu.grace@yahoo.com.tw; 6Department of Physical Therapy, Fooyin University, Kaohsiung 831, Taiwan; ft083@fy.edu.tw; 7Department of Respiratory Therapy, College of Medicine, Kaohsiung Medical University, Kaohsiung 807, Taiwan; 8Center for Fundamental Science, Kaohsiung Medical University, Kaohsiung 807, Taiwan

**Keywords:** motor development delay, infant head lag, artificial intelligence, remote screen, smartphone

## Abstract

Introduction: Video-based automatic motion analysis has been employed to identify infant motor development delays. To overcome the limitations of lab-recorded images and training datasets, this study aimed to develop an artificial intelligence (AI) model using videos taken by mobile phone to assess infants’ motor skills. Methods: A total of 270 videos of 41 high-risk infants were taken by parents using a mobile device. Based on the Pull to Sit (PTS) levels from the Hammersmith Motor Evaluation, we set motor skills assessments. The videos included 84 level 0, 106 level 1, and 80 level 3 recordings. We used whole-body pose estimation and three-dimensional transformation with a fuzzy-based approach to develop an AI model. The model was trained with two types of vectors: whole-body skeleton and key points with domain knowledge. Results: The average accuracies of the whole-body skeleton and key point models for level 0 were 77.667% and 88.062%, respectively. The Area Under the ROC curve (AUC) of the whole-body skeleton and key point models for level 3 were 96.049% and 94.333% respectively. Conclusions: An AI model with minimal environmental restrictions can provide a family-centered developmental delay screen and enable the remote monitoring of infants requiring intervention.

## 1. Introduction

According to the Global Burden of Disease study in 2016, there were more than 53 million children under 5 years old with developmental disabilities (DDs) that would prevent them from reaching their potential in the cognitive, motor, and social-emotional domains [1]. These children raise concern worldwide because they are at risk of having a shorter life expectancy and lower annual income in adulthood [1,2]. Development delay refers to children who are not developing at the expected rate for their age based on standardized milestones of different developmental domains. The cause of development delay may not always be a DD; it can also be due to self-limiting or constitutional delay in age-specific milestones. Early detection and timely intervention for children with developmental delays is an evidence-based approach to improving their developmental outcomes [3]. However, only 30% to 50% of all children with disabilities are identified before school age, which reveals that there is a gap between identifying developmental delays and providing timely interventions [4].

In developmental domains, motor skills serve as the cornerstone for children to explore and engage with the world around them. Motor skills have a critical impact on other developmental domains, including cognitive, social, and emotional development [5,6]. Motor developmental delay can have cascading effects that lead to lifelong consequences, such as poor learning and academic success [7]. Among different motor skills, the development of gross motor domains for children dramatically improves after birth to one year old. Gross motor delay under one year of age may be the consequence of perinatal complications for high-risk infants, such as preterm birth [8] and early signs of neurobehavioral disorders [9,10]. In addition, DDs, such as cerebral palsy (CP) or neuromuscular disease, usually show early gross motor delay [11].

The gold standard for identifying motor development delay and monitoring intervention progress is to use standardized assessor-administered tests, such as the Bayley Scale of Infant Development [12], Hammersmith Infant Neurological Examination (HINE) [13], general movement assessment (GMA) [14], and Alberta Infants Motor Scale (AIMS) [15]. These tests require senior pediatric physical therapists and pediatricians (hereafter referred to as assessors) to conduct at least a 20 min assessment. The time-consuming nature of these tests and the high cost of assessment fees make these tests impractical for universal use [16]. In addition, parent-report questionnaires are an alternative to assessing children’s motor development. These questionnaires are cost-effective and are without temporal and spatial limitations [17]. However, most questionnaires are designed to screen for all developmental domains, not just early motor development. Additionally, there is a risk that parents may overestimate or underestimate their child’s abilities [18,19].

In recent years, as the development of computer vision and artificial intelligence (AI) algorithms has accelerated, there is increasing evidence that technology-assisted assessments of movement in infants’ motor development are possible [20]. In this field, autoaugmentation of the general movement assessment (GMA) has been most studied. The clinical GMA, which assesses the quality of infants’ spontaneous movement in the supine position at 3 to 5 months of age, is highly sensitive to predicting cerebral palsy (CP) [21]. However, there are some challenges that limit the application of these studies to motor development delay detection in the real world. Most studies of GMA have used 3D cameras or sensor-based technology, and few studies have used 2D images analyzed with the optical flow method [22]. Only one study has shown the feasibility of using smartphone images taken by parents to fit a pretrained model from a multicamera setup [23].

AI algorithms that infer human body poses from 2D images [24] and deep learning techniques [25] are promising for this task. However, it is not yet possible to collect enough infant pose data to sufficiently train a deep learning network in the field of infant movement detection [22]. In addition to GMA, other assessments, such as HINE and AIMS, have also shown good predictions for infants’ gross motor delay [26,27]. However, most standardized tests in the early infant stage require poses to be assessed from different visual angles, such as supine, prone, sitting, and standing positions. This can be a challenge for learning algorithms [28].

Considering that more than half of children have transient or reversible developmental delays, studies on applying telehealth and AI-aided systems to improve the care of children with DDs for outpatient screening, diagnosis, intervention, and follow-up are still few, but they should be beneficial [29,30]. Even in low- and middle-income countries (LMICs), mobile phones are game-changers for healthcare, such as in the care of human immunodeficiency virus (HIV) and asthma [31]. The urgent need for family-centered services for children with disabilities is growing, especially in LMICs [32]. Mobile devices are cost-effective, portable, and user-friendly in clinical settings. This paper aims to propose a preprocessing method for mobile-based video from real-world images to identify infants’ head lag without limitations on filming angles [23,28], multicamera settings, special cameras, or training data size [22].

The experimental results of our research demonstrated the possibility of expanding expert classification items from real-world images of younger infants captured by mobile devices using AI methods. Sustainable Development Goal (SDG) 4.2 from the United Nations calls for action to track progress toward early childhood development for children under five years old [32]. Considering that more than half of children with transient or reversible developmental delays are not usually associated with a chronic disease or diagnosis, developing an AI-aided system will benefit primary care providers and families by providing screening, diagnosis, and rehabilitation services [30].

## 2. Materials and Methods

### 2.1. Study Design

The study design is a prospective analysis of videos recorded by parents during follow-up in clinics in the Department of Pediatrics and Neonatology, Kaohsiung Medical University Hospital (KMUH). All the parents of participating infants signed and provided written informed consent, and the principles of the 1964 Declaration of Helsinki and Personal Information Protection Act in Taiwan were followed. The Institutional Review Board of the Kaohsiung Medical University Hospital approved this study (Institutional Review Board number KMUHIRB-SV(I)-20200015).

### 2.2. Data Collection and Measurement

Infants who were discharged from KMUH NICU (Neonatal Intensive Care Unit) with the need for an early intervention program in the high-risk following-up clinics met the inclusion criteria. A total of 41 infants were included from October 2020 to May 2021. For safety reasons, children with home oxygen supply were not included in this study. In the clinics, the 41 infants received the pull-to-sit (PTS) test as a regular neurological examination for head lag. Each examination was evaluated by a senior physical therapist (PT) during their visits. This paper uses several studies [28,33] as a reference and categorizes PTS results into levels 0, 1, and 3, as defined in the first edition of the Hammersmith Infant Neurological Examination (HINE). The interval of motor development assessment for each infant is approximately one to three months as the clinical needs dictate. Two senior PTs, T. Z. Huang and L. C. Chen, were responsible for all head lag scoring in our study. Our PTs, blinded to the infants’ ages, evaluated all 270 randomly ordered videos taken from 41 infants and labeled each one according to the PTS level using HINE criteria. The interrater agreement of the two PTs was excellent (Cohen’s kappa κ = 0.89). The intra-assessor reliability for rating the PTS levels of 27 randomly chosen videos (10% of the sample) was Cohen’s kappa κ = 0.85 for assessor 1 and κ = 0.90 for assessor 2. In the high-risk follow-up clinics, the videos were recorded by parents with a mobile phone from the lateral and foot sides as the infants were pulled from lying to sitting positions by the PTs. The only instruction given to the parents was that the infant’s face and shoulders should be visible in the film. A video was considered invalid if the child was unable to complete the evaluation due to excessive crying or if their face and shoulders were obstructed for more than half of the pull-to-sit duration. Accordingly, 14 of the 284 videos were excluded from our data.

### 2.3. Data Analysis

The process of the proposed method is shown in Figure 1. It consists of three main stages: data acquisition, data preprocessing, and data fuzzy learning. The proposed model is an instance-based learning approach in which the distances between testing instances and training instances are computed based on the fuzzy membership functions derived from data fuzzy learning. In the following subsections, we introduce the details of the proposed method based on Figure 1.

#### 2.3.1. Data Acquisition

The PTS level labels for 270 video annotations were given by assessors based on the criteria described in the first edition of the HINE. To transform infant videos into computable real numbers, human pose estimation (HPE) algorithms were used to infer the skeleton key points of the infants in the videos. In this study, an extension of the Deep-Dual-Consecutive Network (DcPose) [34] was used to automatically track infant actions in videos. The choice of DcPose was based on its ability to make better body key points for video sequences [34]. This study adopted thirteen key points defined in DcPose, including the head, left shoulder, right shoulder, left elbow, right- elbow, left wrist, right wrist, left pelvis, right pelvis, left knee, right knee, left ankle, and right ankle. In addition, we also adopted a five skeleton key point model, including the head, left shoulder, right shoulder, left pelvis, and right pelvis. The choice of DcPose here made better body key-points for video sequences [34].

#### 2.3.2. Data Preprocessing

The objective of data preprocessing is to ensure data correctness for our learning models. There are three issues in learning skeleton data between annotated videos:Different frame numbers (i.e., video duration)Different video visual anglesDifferent infant positions in videos (i.e., coordinates)

To tackle these issues, five steps are proposed in the following paragraphs.

In most cases, the duration of annotated videos is different. This difference can make learning models less robust, especially when there are not enough training samples to train a deep neural network. Since the actions at the beginning (e.g., lying) and end (e.g., sitting) of annotated videos are the same, as shown in Figure 2, we only need to consider the action and its key point positions at a single frame. Therefore, we can set a maximum frame number *N* to limit the total number of frames in annotated videos. Usually, 1 s of video consists of 30 image frames, and we will obtain 30 sets of skeleton key points after DcPose is employed. Taking Figure 2 as an example, when the durations of the three videos are 3 and 6 s, we have 90 and 180 frames, respectively. We can set *N* as 90, the minimum number in the videos, to eliminate some or all even-numbered frames of the other two videos to make all video frames the same.

In some cases, video filming angles can be casual. For example, the PTS criteria described in a previous study state that the head of a baby should follow the torso within 15° for a score of 3 when viewed from the side. However, if a video is recorded from a different angle (e.g., 60°), the 15° criterion should be adjusted accordingly [29]. It is challenging for assessors to accurately score infant actions from different visual angles, and the same is true for learning algorithms. To address this issue, 2D key points need to be rotated to the side view. This can be achieved in three steps: 3D key point transformation, 3D key point rotation, and 2D key point projection.

To transform 2D key points into 3D key points, this paper adopted the cascaded deep monocular 3D HPE (referred to as EvoSkeleton) [35]. Rotating 3D key points into an expected visual angle is a customized task according to the criteria defined in assessments. For example, for PTS, the *XY* plane is rotated to make the torso approximately parallel to the horizontal (*Y*-axis). Then, the *XZ* and *YZ* planes are rotated to align the left-shoulder key point with the right-shoulder key point and the left pelvis key point with the right pelvis key point. For a given 2D coordinate (*x*, *y*), its rotated coordinate (*x′*, *y′*) can be computed by
(1)$$\left[\begin{array}{c} x^{\prime }\\ y^{\prime } \end{array}\right]=\left[\begin{array}{c} x\;\cos\theta -y\;\sin\theta \\ x\;\sin\theta +y\;\cos\theta \end{array}\right]$$
where θ is the angle. After the rotation process, we project the 3D skeleton keypoints onto the *XY* plane to retrieve 2D skeleton key points. This can be accomplished by removing the Z-axis data. Note that it is not necessary to perform the projection if the assessments provide 3D criteria.

Finally, considering that infant position and size could be quite different between videos, a normalization operation is needed to standardize the *X* and *Y* coordinates of skeleton key points into [0, 1] by
(2)$$\left[\begin{array}{c} x^{\prime }\\ y^{\prime } \end{array}\right]=\left[\begin{array}{c} \left(x-min\left(X\right)\right)/\left(\max\left(X\right)-\min\left(X\right)\right)\\ (y-min(Y))/\left(\max\left(Y\right)-\min\left(Y\right)\right) \end{array}\right]$$
where $min\left(X\right)$, $\max\left(X\right)$, min(Y), and $\max\left(Y\right)$ are the minimum and maximum values of *X* and *Y* coordinates in skeleton key points in the whole annotated frames.

#### 2.3.3. Data Fuzzifying Learning

When the training sample size is small, it is often not possible for neural networks to derive robust predictions because of an insufficient training process [22]. To overcome this, and based on the proposed data preprocess, we developed an instance-based learning approach in which distances between independent testing instances and training instances are calculated using fuzzy techniques. With the fuzzy instance-based approach, label possibilities for a given series of preprocessed skeleton data can be systematically inferred. Suppose that there are 30 infant videos. After our data preprocessing, the frame size is a fixed number, all *X* and *Y* coordinates have been standardized into the range [0, 1], and the angles of bodies have been rotated into a standardized view. Here, we take the 30 head key points at the first frame as an example, as shown in Figure 3. According to the definition in statistics, 99.73% of the data should be within three standard deviations from their averages. Therefore, we can infer the lower bound (LB) and the upper bound (UB) of a group of observations by
(3)$$\left\{\begin{array}{c} \mathrm{LB}=\mathrm{Avg}-3\times \mathrm{Sd}\\ \mathrm{UB}=\mathrm{Avg}+3\times \mathrm{Sd} \end{array}\right.$$
where Avg and Sd denote the average and the standard deviation of observations, respectively. Based on LB, Avg, and UB, we can construct a fuzzy triangular member function (MF) as
(4)$$MF(x)=\left\{\begin{array}{c} \left(x-LB\right)/\left(Avg-LB\right),\;x<\mathrm{Avg}\\ \left(UB-x\right)/\left(UB-Avg\right),\;x\geq \mathrm{Avg}\\ 0,\;\mathrm{otherwise} \end{array}\right.$$

The MF value, MF(*x*), is regarded as the plausibility (or said possibility) of *x* in fuzzy theories. In Figure 3, we can see that the two MFs are constructed based on the 30 head key point coordinates.

Suppose that the frame number is *N* and the skeleton key point number is *k*. We will have N×k×2 triangular MFs after data fuzzification for training data, where 2 is the *X* and *Y* axes. We call these MFs the prior knowledge, which was extracted from the training data. Based on the mechanism of instance-based learning, when an independent test instance is given, we can compute the possibility of a key point at a frame by using the geometric means as
(5)$$poss=\sqrt[{2}]{{MF}_{X}(x)\times {MF}_{Y}(y)}$$
where ${MF}_{X}$ and ${MF}_{Y}$ are the keypoint’s *X* and *Y* MFs, respectively. The possibility for a frame (FramePoss) is the average calculated by
(6)$$FramePoss=\sum_{i=1}^{k}{poss}_{i}/k$$

The final possibility (*fp*) of the independent test instance is the average computed as
(7)$$fp=\sum_{i=1}^{N}{FramePoss}_{i}/N$$

Note that the independent test instance has been processed by our data preprocessing. We use *fp* to evaluate how close the independent test instance is to the prior knowledge, i.e., the distances between testing instances and training instances in instance-based learning. When most of the key point positions are not in the MF, *fp* is very small because most of the MF values are zero. When data have multiple labels, we need to repeat this data fuzzification for each label to extract multiple sets of prior knowledge. In this study, fuzzy models of levels 0, 1, and 3 will be built first, and the label of an independent test instance will be determined by its maximum *fp* in the three models. For example, if the maximum *fp* comes from level 0, then the instance will be classified as 0.

### 2.4. Empirical Evaluation

#### 2.4.1. Experiment Design

To obtain a robust experimental result, we conducted 30 repeated five-fold stratified cross-validations. In stratified cross-validation, the label distribution rates of the training and testing sets will remain approximately the same. In addition to the thirteen skeleton key points provided by DcPose, we selected five skeleton key points that are considered more relevant to detecting head lag for an effectiveness comparison. The five key points are the head, right shoulder, left shoulder, right pelvis, and left pelvis.

#### 2.4.2. Evaluation Metrics

The experiment was implemented in Python 3.7 code. The details of the Python environments for DcPose [34] and EvoSkeleton [35] can be found on their GitHub websites. The mission of this experiment is to identify the PTS scores of infants. Therefore, this study adopted accuracy, sensitivity, and specificity as the evaluation indicators for our experiments. The indicators are computed by
(8)$$accuracy=\frac{TP+TN}{TP+TN+FP+FN}$$
(9)$$sensitivity=\frac{TP}{TP+FN}$$
(10)$$specificity=\frac{TN}{FP+TN}$$
where *TP*, *TN*, *FP*, and *FN* denote true positives, true negatives, false positives, and false negatives, respectively. In addition, kappa is also adopted as an indicator to test the interrater reliability between the PTs and our models. Kappa is computed using the Python function “cohen_kappa_score” in the sklearn package.

To identify whether significant differences exist between the two models built using thirteen key points and five key points with statistical support, paired *t* tests with a two-tailed test are performed. The null hypothesis ($H_{0}$) and alternative hypothesis ($H_{a}$) are defined as
(11)$$\left\{\begin{array}{c} H_{0}\;:\mu_{d}=0\\ H_{a}\;:\mu_{d}\neq 0 \end{array}\right.$$
where $\mu_{d}$ is the average of $\left\{\;d_{j}\left|j=1,2,\ldots ,k\;\right.\right\}$ and $d_{j}$ is the deviation between the two accuracy indicators of the two models in the *j*th cross-validation run.

Since PTS level labels are an ordinal scale rather than a nominal scale, the above metrics are computed in a binary method when experiments are performed. That is, we take the predicted label of an independent test instance as level 1 when our models give labels 1 or 2 to discriminate the level degree between 0 and those greater than or equal to 1. Next, the predicted label is regarded as 0 when our models output labels 0 or 1.

### 2.5. Implementation Details

The details of the implementation steps in the proposed method are given as following steps. Note that the PTS level labels of the videos have been annotated by PTs.

Step 1. Python’s OpenCV package, named cv2, is used to capture all videos into image frames.Step 2. The video frame numbers are standardized by removing some or all even image frames, where the standard number is the minimum frame number in all videos.Step 3. The Deep-Dual-Consecutive Network (DcPose) [34] is employed to retrieve 2D skeleton key points of infants from image frames.Step 4. EvoSkeleton [35] is adopted to infer the 3D skeleton key points of the 2D key points.Step 5. The 3D skeleton key points ae rotated into the required visual angles for an assessment using Equation (1).Step 6. The rotated 3D skeleton key points are projected onto the *XY* plane to retrieve new 2D skeleton key points by removing the Z-axis data.Step 7. The *x* and *y* coordinates of the new 2D skeleton key points are normalized into 0 and 1 using Equation (2).

The above seven steps are data preprocessing. Next, the videos are separated into a training set and a testing set using stratified-fold cross-validation. The training process is described as follows:Step 8. The training set is separated into three subsets according to the PTS levels 0, 1, and 3.Step 9. For each PTS level label, *x* and *y* fuzzy MFs are built for thirteen skeleton key points using Equations (3) and (4). This will result in three PTS level models, each of which has 26 MF sets.

After Step 9, the model training is complete. The steps in the testing process are described as follows:Step 10. For each testing instance, the final possibilities (*fp*s) are computed in the three PTS level models using Equations (5)–(7).Step 11. The TPS level labels of the testing instances are determined by their maximum *fp*.Step 12. The accuracy, sensitivity, specificity, and kappa evaluation metrics for the testing result are calculated using Equations (8)–(10) and the “cohen_kappa_score” function in the “sklearn” package.Step 13. Steps 8 to 12 are repeated until a five-fold stratified cross-validation is performed.

## 3. Results

In our data, the profiles of 41 infants are summarized in Table 1. Thirty-seven of the 41 infants were preterm infants. The mean birth body weight was 1735 g, and the mean birth gestational age was 31.7 weeks. The mean age while filming was a corrected age of 4.8 months old. There was 1 infant diagnosed with cerebral palsy and 7 infants with gross development delay status. A total of 270 videos were collected for analysis, and 84, 106, and 80 videos were labeled “level 0,” “level 1,” and “level 3,” respectively.

The experimental results regarding accuracy are listed in Table 2. When the label is level 0 vs. levels 1, 3, the averaged accuracies of using thirteen skeleton key points and five key points are 77.667% and 88.062%, respectively. When the label is levels 0, 1 vs. level 3, the averaged accuracies of using thirteen skeleton key points and five key points are 96.049% and 94.333%, respectively. From Table 2, we see the following results: Adopting five key points is better than using thirteen key points with statistical support (*p* value is 4.84 × 10^−24^) when the label is level 0 vs. level 1. However, the situation is reversed when the label is levels 0, 1 vs. level 3 (*p* value is 6.99 × 10^−13^). In other words, using five key points is more accurate for distinguishing between levels 0 and 1, while using thirteen key points is more accurate for distinguishing between levels 0, 1, and 3. On the whole, taking five key points to build our models can represent a more robust testing result than using thirteen key points.

Confusion matrices of the 30 repeated five-fold stratified cross-validations are summarized in Table 3. In each five-fold stratified cross-validation, 270 instances were tested in turn. Accordingly, there are 8100 testing results. Based on Table 3, specificity and sensitivity metrics are computed and listed in Table 4, where “Averages” are the averages of specificity and sensitivity. Table 2 and Table 3 show that our models have high effectiveness when the original video data have been well treated by the proposed preprocessing. In addition to the accuracy, specificity, and sensitivity metrics, the kappa values are summarized in Table 5. The kappa values are all greater than 0.5 and reach almost perfect agreement when our models are taken to distinguish between levels 0, 1 vs. level 3.

## 4. Discussion

Our study provides a simple, straightforward pipeline for computer-based PTS labeling. With the AI-aided autotracking process and the proposed data preprocessing, infants’ skeletons can be captured on mobile videos without background and angle limitations and provide features for model recognition. A total of 42.4% of the preterm infants had head lag until term equilibrated age, and only 4% of the infants had the same symptoms at birth [8]. While head lag has proven to be a vital early warning sign and PTS is an easy and necessary neurologic examination, few studies have explored how technology could help identify it. Dogra DP et al. [28] first reported the autorecognition program by a preprocessing algorithm for PTS leveling, and they needed infants dressed off with two cameras and a fixed distance in the lab. The sensitivity and specificity were 80% and 89%, respectively, in the marker-free group and more than 95% when using the body marker. With the same research teams and video sets, they tried to autorecognition body features in a video frame by frame through the scale-invariant-feature transform descriptor with a machine learning classifier, and the accuracy of PTS leveling was up to 84% [33]. In our study, the performance of the five skeleton key points was superior to the whole-body skeleton points model on the whole. According to the grading criteria of Hammersmith, there is a difference in the angle between the central axis and the head between level 0 and level 1. However, at the beginning of the movement, they may appear similar, and the head is generally unable to align with the central axis. Therefore, the currently limited sample size and the lack of angle restrictions contributed to the indistinguishable results between the two.

Traditionally, the criteria defined in HINE for the two levels are whether the angle between the head and torso exceeds 30°, and the leg is not the primary basis for judgment. In our data, some level 0 infants presented an angle very close to 30°. As defined in Equation (3), the estimation of data bounds results in a more significant overlap between the critical points of levels 0 and 1, especially in the leg skeleton key points. Therefore, the key points of infants’ legs in PTS items become noise data that lower discrimination capability. Traditionally, researchers use markers to improve computer vision to identify important body parts, such as heads, shoulders, and pelvis, for PTS classification [28]. The better result from five points reflected that putting domain knowledge into neural networks helps improve discrimination, as in other studies and small study sets [35].

Our contribution and strength are that we are the first to prove that 3D skeletons can be appropriately aligned from mobile 2D images with tracking, prediction skeletons, and model building for classification items of standardized examiner-administered tests in infants’ motor development. The traditional burden of methodologies while analyzing movement with 2D video is that a singular vantage point may impede the body segments and make the distance unmeasurable. The 2D open pose-based human skeleton method recorded in the lab to classify GMA has been proven to have a major advantage over state-of-the-art ML-driven methods [36]. The feasibility of GMA video recording by parents with smartphones with background and angle limitations at home has proven acceptable for remote assessment by preprocessing automated scoring [37]. J Zhou et al. combined a CNN and a hierarchical 3D pose estimation scheme to recognize the item of AIMS, but they used synthetic infant images to train and validate their models [38]. For preschool children, a labor-saving AI assessment system, combined with tracking process and gross motor action recognition such as jumping and running with CNN and 2D skeleton, demonstrated an accuracy of 82.3% on daily activity videos [39]. Despite the promising and growing application of telemedicine in medicine, its application in pediatric and neonatal settings is limited [40]. In addition, current AI applications for pediatric rehabilitation focus on delivery services, but gaps in personalization approaches and remote real-time evaluation still exist [41].

## 5. Limitations

Our study had several limitations. First, we only validated the autorecognition of PTS. The prevalence of motor delay ranged from 2.3% to 5.5%, and another 8.8% of patients who were not on track needed close monitoring [4]. We chose PTS as an example for proof of concept because it is one of the essential items from HINE and an easy-to-use neurologic examination in clinics. Although head lag alone could not predict future NDI outcomes [8], infants with CP or neuromuscular disease usually showed early head lag [11]. Compared with low-risk infants, head lag was demonstrated as an early predictive biomarker of future communication delays and diagnosis of autism in children with autistic siblings [9,10].

Our study could inspire further research on the monitoring of more complicated combinations of motor skills with smartphone devices. Based on the skeleton key points derived by HPE algorithms [34], our results demonstrated the feasibility of applying the proposed AI skeleton and fuzzy learning method to medical screens. However, there are two limitations when using our method. One is that our fuzzy learning method can only achieve expected results with the key point data transformed by our data preprocessing, and the other is the incorrect key point coordinates output by HPE algorithms. HPE algorithms sometimes output incorrect keypoint coordinates, and researchers are still working on solving this issue [34]. Although there are some open datasets for researchers to develop action-recognizing models, most of these datasets are used for learning everyday actions, such as handshaking, talking, standing, and sitting, and do not apply to infant head lag studies [42]. In addition to calling for more open datasets in this field, further studies could explore the potential of the generative adversarial network (GAN) to influence the technology-assisted diagnosis of developmental delays in infants [21].

Finally, our sample was obtained from a high-risk neonatal tracking program. Although the assessment criteria for motor development in children are consistent, regardless of whether they are typically developing infants or high-risk infants, we still cannot determine the performance of our model on low-risk newborns. Therefore, further samples and studies are needed to validate this aspect.

## 6. Conclusions

In this paper, we use two neural network architectures, DcPose and EvoSkeleton, and an instance-based fuzzy model to predict the infant skeleton and perform the classification of the PTS test through subtle changes in the skeleton without distance limitations, clothing limitations, or occlusion. We proved that by using videos captured on a mobile device with minimal limitations and AI methods, the accuracy of recognizing PTS levels based on HINE is not inferior to previous automatic recognition systems under restricted filming environments. Our research demonstrates the possibility of expanding expert classification items from real-world images of younger infants captured by mobile devices using AI methods. Further research on the automatic recognition of infant motor developmental delays should focus on standardized examination results to facilitate early diagnosis and remote rehabilitation applications.

## Figures and Tables

**Figure 1 children-10-01239-f001:**
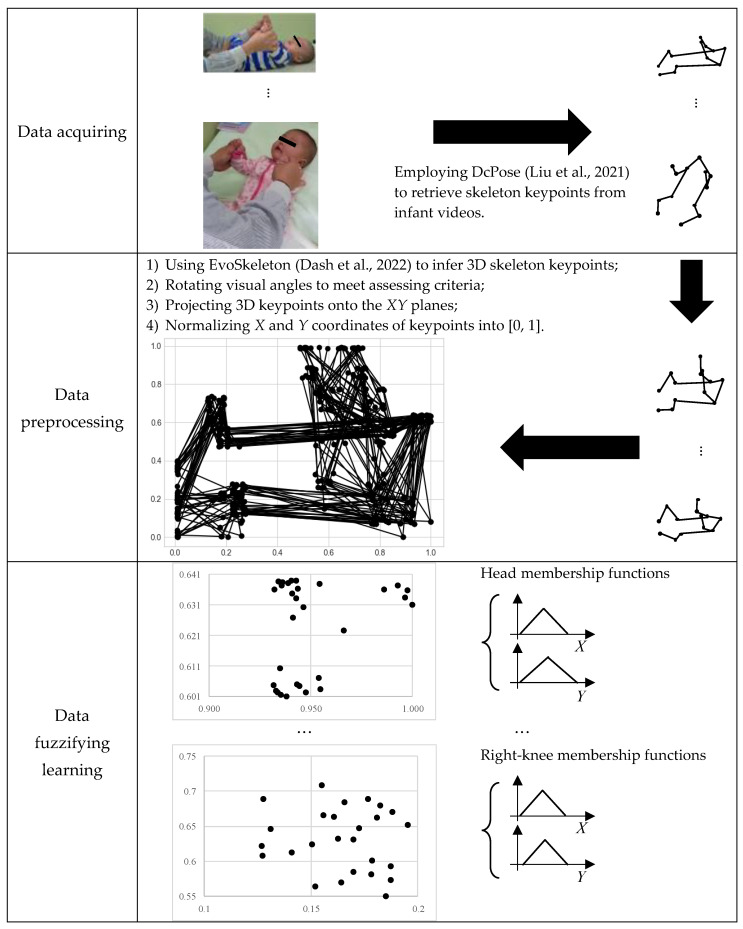
The process of the proposed method [34,35].

**Figure 2 children-10-01239-f002:**
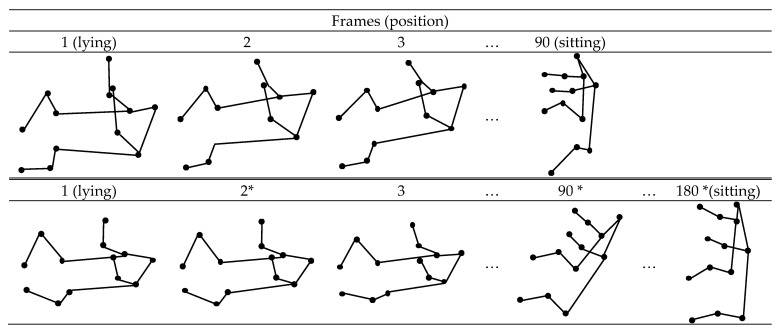
Diagram of two videos with different durations. * indicates that the frames need to be eliminated.

**Figure 3 children-10-01239-f003:**
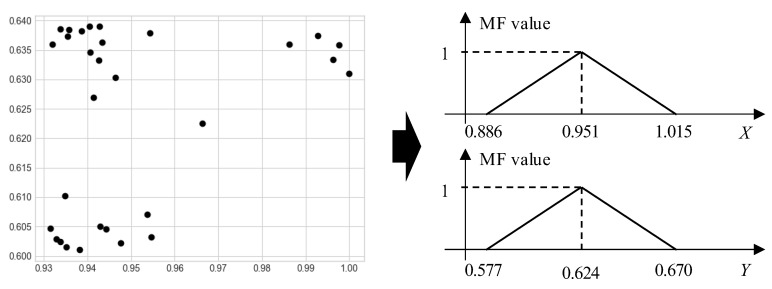
Data fuzzification for *X* and *Y* values of 30 head key points at a frame.

**Table 1 children-10-01239-t001:** Infant profiles in our collected data.

	Participate (*n* = 41)
Gender, *n* (%)	
Boy/Girl	22 (53.6)/19 (46.4)
GA, wk, mean ± SD	31.7 (3.9)
Birth weight, g, mean ± SD	1735.5 (704.8)
Type of delivery, C/S, *n* (%)	24 (55.8)
Preterm, *n* (%)	37 (90.2)
Meningitis, *n* (%)	4 (9.8)
Preeclampsia, *n* (%)	7(17.1)
GDM, *n* (%)	5 (12.2)
Maternal age, years, mean ± SD	34.2 (4.2)
Maternal education < 12 years, *n* (%)	14 (34.1)
PVL, *n* (%)	1 (2.4)
IVH, *n* (%)	9 (21.9)
BPD, *n* (%)	12 (29.3)
NEC, *n* (%)	0 (0)
Corrected age while recording, months, mean ± SD	4.8 (2.9)
Gross Motor Development Delay, *n* (%)	7 (17.0)
Cerebral Palsy, *n* (%)	1 (2.4)
HINE PTS, *n*	270
Level 0, *n* (%)	84
Level 1, *n* (%)	106
Level 3, *n* (%)	80

GA: gestational age; SD: standard deviation; C/S: Caesarean section.

**Table 2 children-10-01239-t002:** Averages (Avgs) and standard deviations (SDs) of accuracy and their p values of two-tailed paired *t* tests in the 30 repeated five-fold stratified cross-validations.

Label	Level 0 vs. Levels 1, 3		*p*-Value
Key Points	Thirteen	Five	
Avgs	77.667%	88.062%	4.84 × 10^−24^
SDs	1.265%	1.225%	
**Label**	**Levels 0, 1 vs. level 3**		***p*-Value**
**Key Points**	**Thirteen**	**Five**	
Avgs	96.049%	94.333%	6.99 × 10^−13^
SDs	0.633%	0.468%	

**Table 3 children-10-01239-t003:** Confusion matrices of the thirty repeated five-fold stratified cross-validations.

		Predicted Labels			
		Thirteen Key Points		Five Key Points	
		Level 0	Levels 1, 3	Level 0	Levels 1, 3
True labels	Level 0	2498	22	2468	52
	Levels 1, 3	1787	3793	915	4665
		Levels 0, 1	Level 3	Levels 0, 1	Level 3
	Levels 0, 1	5624	76	5588	112
	Level 3	244	2156	347	2053

**Table 4 children-10-01239-t004:** Results of specificity and sensitivity of the 30 repeated five-fold stratified cross-validations.

	Thirteen Key Points		Five Key Points		Supported Instances
	Specificity	Sensitivity	Specificity	Sensitivity	
Level 0	99.127%	99.127%	97.937%	83.602%	2520
Levels 1, 3	67.975%	67.975%	83.602%	97.937%	5580
Averages	83.551%	83.551%	90.769%	90.769%	
Levels 0, 1	98.667%	89.833%	98.035%	85.542%	5700
Level 3	89.833%	98.667%	85.542%	98.035%	2400
Averages	94.250%	94.250%	91.788%	91.788%	

**Table 5 children-10-01239-t005:** The kappa values of the 30 repeated five-fold stratified cross-validations.

Labels	Thirteen Key Points	Five Key Points
Level 0 vs. levels 1, 3	0.563 ^1^	0.745 ^2^
Levels 0, 1 vs. level 3	0.903 ^3^	0.860 ^3^

^1^: moderate agreement; ^2^: substantial agreement; ^3^: almost perfect agreement.

## Data Availability

The data cannot be shared because the initial informed consent ensured privacy.

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
