# Peer review of "Mobile Device-Based Video Screening for Infant Head Lag: An Exploratory Study"

_children, 2023, doi:10.3390/children10071239_

Round 1
Reviewer 1 Report
I would, first of all, like to congratulate the authors on the great work and wonderful idea with this article. Prior publication, there are some issues that should be resolved.
1. In the line 279 of the discussion section, the authors state the differences in the sensitivity and specificity of the 13 key points approach between the level 0 and level 1 infants, would be great to have a possible explanation for this?
2. Line 282, 'parts of level 0 infants', it is some of level 0 infants, correct?
3. There is a typo in line 289, should be: 'improve'
4. Lines 310 to 315 are better for the introduction
Reviewer 2 Report
The present study used phone recorded videos of infant movement to improve upon use of photos for AI training to detect motor delay and difficulties in infants. The videos were with infants with high-risk of motor delay, mainly associated with prematurity. The research provides a valuable starting point for use of AI in diagnostics. I have two main points for consideration.
1. There is a focus on diagnostics, but I wondered if the potential use is more likely to be in screening.
2. The present study has been conducted with a high risk clinical sample. The emphasis in the introduction (e.g. lines 33-44) and parts of the discussion seems to be all children who may have difficult to detect signs of disability. While it can be helpful to aspire to development of tools that will support early detection of disability in all children, at the moment the focus should be on children already identified as high-risk.
Thank you for the opportunity to review this interesting study.
Copy-editing is needed. There are a few sentences that don't make sense and overall the article is difficult to read due to error is in English expression.
Reviewer 3 Report
Automation of developmental motor assessments is an important, active field, and this work extends a growing toolkit and body of evidence. Clarification of methodology is needed to permit replication and to assess the generalizability of results.
Title: This study examines only one item of the HINE-- i.e., head lag in pull-to-sit. The title of the article should reflect this narrow scope.
Introduction:
-Authors should revisit discussion of using machine learning-based tools in infant motor assessments and this study's place within the development of such a toolkit in Paragraph 4. I agree that machine learning-based tools, in principle, have the potential to automate/facilitate currently-available clinical assessments and even to extend/augment them (e.g., by adding precision; by capturing nuances that are difficult for humans to see). I also agree with the authors' approach of attempting to automate an existing measure (here, within a classification scheme) before attempting to extend/augment existing tools. However, Introduction paragraphs 4-6 should explicitly clarify this study's Aims within this framework.
-Prior art should be discussed in greater detail. Which elements of this processing pipeline have been used in similar classification tasks, and which are novel?
-Minor point: I don't necessarily agree that privacy and security concerns are the main limitation preventing the development of machine learning-based assessment tools. As is the case for much of the machine learning field, a major limitation is the need for large, well-labeled datasets-- datasets that are expensive and time-consuming to produce.
Methods:
-2.2.: Please explicitly describe inclusion and exclusion criteria. Please also expand upon logistics: were parents responsible both for performing the pull-to-sit maneuver as well as for recording? Was clinical assessment also done at the visit for validation? Were any videos obtained and submitted that were not of suitable quality for rating?
-Sections 2.3.x contain substantial text regarding conceptual frameworks but do not contain enough detail regarding actual processing steps. Please include enough detail to permit replication (including and up to code availability if applicable).
-Please clarify in particular regarding the novelty vs. prior art regarding "data fuzzifying learning." If I understand correctly, it seems to resemble a perceptron as used in most neural networks? Authors appear to non-linearly transform observations (using a "fuzzy triangular member function"; essentially normalizing weights) then integrate them (weighted sum)
-I do not understand how the time dimension is used in analyses
-Figure 2 does not currently aid understanding
-I do not understand how to interpret the authors' apparent application of multiclass ROC statistics. Classical ROC statistics (e.g. TP, TN, FP, FN, AUC) apply only to binary classifications; as there are 3 classification outcomes in this case (0, 1, and 3), the authors appear to use a multiclass generalization. The methods referenced seem to permit "one-vs-one" or "one-vs-rest" binarizations of multiclass data. If I understand correctly, authors used "one-vs-rest" methdology. Regardless, this seems to treat classification levels as nominal rather than ordinal-- 3 > 1 > 0. I recommend using ordinary ROC statistics-- but binarizing using cutoffs. That is, compute sensitivity, specificity, and AUC for each of two analyses: 1) classification accuracy of level<0.5 vs. level>0.5 and 2) classification accuracy of level>2 vs. level<2
-Authors should report kappa in addition to raw accuracy to describe AI vs. PT classification accuracy
Results:
-Participant characteristics table: In addition to perinatal characteristics, current characteristics are also important (in particular, age-- whether chronological, adjusted, or both). Current medical and developmental status would also be helpful to the degree available.
-Listing rows for each iteration is not helpful. Summary statistics of the distributions of results (e.g., accuracy, kappa, sensitivity for score<0.5, specificity for score<0.5, AUC at 0.5 cutoff, etc), would suffice (e.g. mean and standard deviation)
-Authors claim performance differences between analysis performed using 13 vs. 5 key points-- statistics should explicitly evaluate these differences
Discussion/Limitations/Conclusions:
-My comments regarding the Introduction apply here also
Please proofread-- many minor errors and inaccuracies
Round 2
Reviewer 3 Report
I appreciate Authors' substantial revisions and find the manuscript substantially improved. Authors' clarifications (particularly regarding their Methods and Results) give me a much better understanding of the study and its findings. However, they also raise serious follow-up questions.
Point/response 3: Please further clarify-- were all infants meeting criteria enrolled, i.e., was enrollment consecutive?
Point/response 4: Much clearer-- thank you. Minor comment-- the frame number normalization process (Step 2) strikes me as atypical and likely to introduce artifact (e.g., altering apparent velocities in non-linear ways). This is not necessary to address in this manuscript but may be an opportunity for improvement in the future.
But critically (and leading into point/response 6): I still do not understand how time dimension handling makes sense. Intrinsically, the head lag procedure is dynamic-- head lag is present when, over the course of being pulled to sit, the neck assumes an extensor position. However, Equation 7 instead seems to average over all timepoints-- in effect deleting all dynamic spatial/temporal information. It would seem that your classifier is essentially making decisions based on a fuzzy, weighted mean position for each keypoint. This would seem to lack physical and physiologic interpretation and would suggest that high classifier performance is instead epiphenomenal-- relying, for instance, on differences in video acquisition positioning in children with head lag. Please address this concern before further review can be considered.
